

# Autophagy-related genes prognosis signature as potential predictive markers for immunotherapy in hepatocellular carcinoma

Deli Mao*, Zhe Zhang*, Xin Zhao and Xiaoqiang Dong

Department of General Surgery, The First Affiliated Hospital of Soochow University, Suzhou, China
* These authors contributed equally to this work.

## ABSTRACT

Autophagy-related genes (ATGs) depress tumorigenesis. However, in tumor tissue, it promotes tumor progression. Here, we demonstrated that 63 ATGs were differentially expressed in normal tissues and tumor tissues of hepatocellular carcinoma (HCC), and seven prognostic-related genes were chosen to establish prognostic risk signatures. It is not just an independent prognostic factor for HCC, but also closely related to the degree of malignancy of HCC. Further, the hallmarks of PI3K–AKT–mTOR signaling was significantly enriched in the high-risk group. Moreover, AKT–pS473 and mTOR–pS2448 expression was down-regulated and correlated with patient prognosis in high-risk group. Finally, we demonstrate that the prognosis signature of ATGs is closely related to immune cell infiltration and PD-L1 expression. In conclusion, ATGs are a crucial factor in the malignant progression of HCC and will be a new prognostic marker for diagnosis and treatment. ATGs prognostic signatures are potentially useful for predicting PD-L1 therapeutic effects.

## INTRODUCTION

Autophagy is the induction of lysosomal degradation of damaged proteins and organelles. It is a process of recycling metabolic substances, providing important nutrients to cells and reducing cellular stress. In normal tissues, autophagy maintains homeostasis, limits cell proliferation, repairs damaged DNA and maintains genomic integrity to inhibit tumorigenesis (*Mah & Ryan, 2012*). It has been previously reported that the autophagy gene Beclin 1 is a tumor suppressor gene. The deletion rate of Beclin 1 gene in breast cancer is 50–70%, and that in ovarian cancer is 75% (*Liang et al., 1999*). In tumor tissues, autophagy has complex biological processes, and different tissue types exhibit different effects. After autophagy in ATG5-deficient mice is suppressed, only liver tissue undergoes neoplastic changes (*Strohecker & White, 2014*). Autophagy mainly manifests in promoting tumor progression in malignant tumor tissues. The study found that the infinite proliferation of tumors is limited by the lack of blood supply. In the tumor hypoxia center,

Corresponding author
Xiaoqiang Dong,
dongxq@hotmail.com

autophagy activity is initiated, relieving metabolic stress, and finally restoring tumor blood supply (*Mathew, Karantza-Wadsworth & White, 2007*). Cumulative evidence indicates that tumor cells remaining in tumor therapy enter dormancy, during which autophagy can promote the survival of dormant tumor cells and escape treatment (*Sosa, Bragado & Aguirre-Ghiso, 2014*). CTLA4 and PD-L1 blockers have been proven to have therapeutic effects in a variety of tumors, but some patients still show no reactivity (*Pietrocola et al., 2017*). This forces us to explore new blockers that depress tumor progression. *Makhov et al. (2014)* found that Piperlongumine inhibits tumor growth and promotes autophagy by inhibiting AKT/mTOR signaling pathway. In prostate cancer, SGK1 promotes autophagy, mTOR phosphorylation increases autophagy activation, and SGK1 and mTOR dual depression have significant anti-tumor effects (*Liu et al., 2017*).

Hepatocellular carcinoma (HCC) is the main primary tumor of the liver. The developing countries in Asia, which are headed by China, are high-risk areas of HCC (*Ghouri, Mian & Rowe, 2017*). HCC often reaches the advanced stage of the disease when it is diagnosed, and the tumor is prone to multi-drug resistance. Although the development of new targeted drugs is used to treat HCC, the overall prognosis is still not improved (*Cabibbo et al., 2010*). Immunotherapy is the Hope of Advanced Tumor (*Ringelhan et al., 2018*), Phase I clinical trials showed that response rates to CTLA4 and PD-L1 Immunological checkpoint inhibitors were 17.6% and 20% (*El-Khoueiry et al., 2017*; *Sangro et al., 2013*). Autophagy is ubiquitous in the HCC process and is the basis of HCC invasiveness, The combination of autophagy inhibitor and immunotherapy is especially useful (*Cui, Gong & Shen, 2013*). Through the combined analysis of TCGA and ICGC databases, we studied the effects of autophagy-related genes (ATGs) on the clinicopathological features and prognosis of HCC and explored the possible signaling pathways involved in autophagy in HCC. Finally, we demonstrate the relationship between ATGs prognosis signature and immune cell infiltration and immunotherapy. To provide new insights for autoimmune inhibitors as potential predictive markers for immunotherapy in HCC.

## MATERIALS AND METHODS

### Datasets

The RNA-seq transcriptome profiling and corresponding clinicopathological information were obtained for 377 HCC patients from The Cancer Genome Atlas (TCGA) (https://portal.gdc.cancer.gov/) and 260 HCC patients from International Cancer Genome Consortium (ICGC) (https://icgc.org/). HCC patient protein expression information and normalized RPPA value of TCGA-LIHC-L3 datasets downloaded from The Cancer Proteome Atlas (TCPA) (https://tcpaportal.org/tcpa/index.html). A total of 232 ATGs were obtained from the Human Autophagy Database (http://www.autophagy.lu/index.html).

### Bioinformatic analysis

Based on the expression of ATGs in normal tissues and tumor tissues of HCC based on the TCGA dataset, we screened 63 differentially expressed genes (logFC > 1 or logFC < −1, and

adjusted *P*-value < 0.05) for later analysis. The Database for Annotation, Visualization and Integrated Discovery (David, V6.8) online website carries out Gene Ontology (GO) and Kyoto Encyclopedia of Genes and Genomes (KEGG) annotation and visualization on differential genes. Univariant Cox regression analyses were used to determine the prognostic value of ATGs in HCC. A total of 21 ATGs were screened and were closely related to HCC prognosis (*P* < 0.001). Finally, we selected seven genes from 21 prognosis-related genes to establish a prognostic risk signature according to the LASSO Cox regression algorithm using the "glmnet" and "survival" R-packages (*Sauerbrei, Royston & Binder, 2007*). The seven genes and corresponding coefficients are established by minimum partial likelihood deviance. The sum of the seven genes and the coefficient product is the risk score for each patient. Based on the median risk score, the TCGA and ICGC HCC patients were divided into high-risk and low-risk groups. Gene set enrichment analysis (GSEA) for high-expression genes in the high-risk group (*Subramanian et al., 2005*). The CIBERSORT method calculates the infiltration abundance of immune cells using the "e1071," "BiocManager" and "parallel" R-packages, which calculates the cell composition according to the complex tissue gene expression profile (*Newman et al., 2015*). The deconvolution approach Tumor IMmune Estimation Resource (TIMER) was used to verify the results (*Li et al., 2016*). In this study, we only revealed macrophage infiltration results.

## Statistical analysis

Unpaired Student's *t*-test was used to compare differences in gene expression between the two groups. The test of correlation coefficients was used to compare the data correlation between the two groups. The TCGA and ICGC datasets HCC patients were divided into high and low risk groups according to the median value of the risk score. The chi-square test was used to compare the clinical-pathological features of HCC between the two risk groups. Univariate and multivariate Cox regression analyses were used to evaluate the prognostic value of risk score and clinicopathological features in patients with HCC. Receiver operating characteristics (ROC) curves test the efficiency of risk score and clinicopathological features in predicting the 5 year survival rate of HCC. The Kaplan–Meier curve with the log-rank test was performed to compare the overall survival (OS) of the HCC patients in two groups. All statistical analyses were conducted using R v3.6.0, SPSS 19.0 and Prism 7.

## RESULTS

### Screening and enrichment analysis of ATGs

We obtained 232 ATGs from Human Autophagy Database. Based on the expression in HCC, 63 differentially expressed genes in the TCGA dataset (logFC > 1 or logFC < −1, adjust *P*-value < 0.05) (Fig. 1A) were screened out, of which FOS, DIRAS3, NRG1, FOXO1 were down-regulated in HCC. The other genes were up-regulated in expression. GO enrichment analysis showed that these genes were mainly enriched in autophagy, the release of cytochrome c from mitochondria and regulation of protein localization to the membrane. KEGG enrichment analysis showed that these genes were mainly

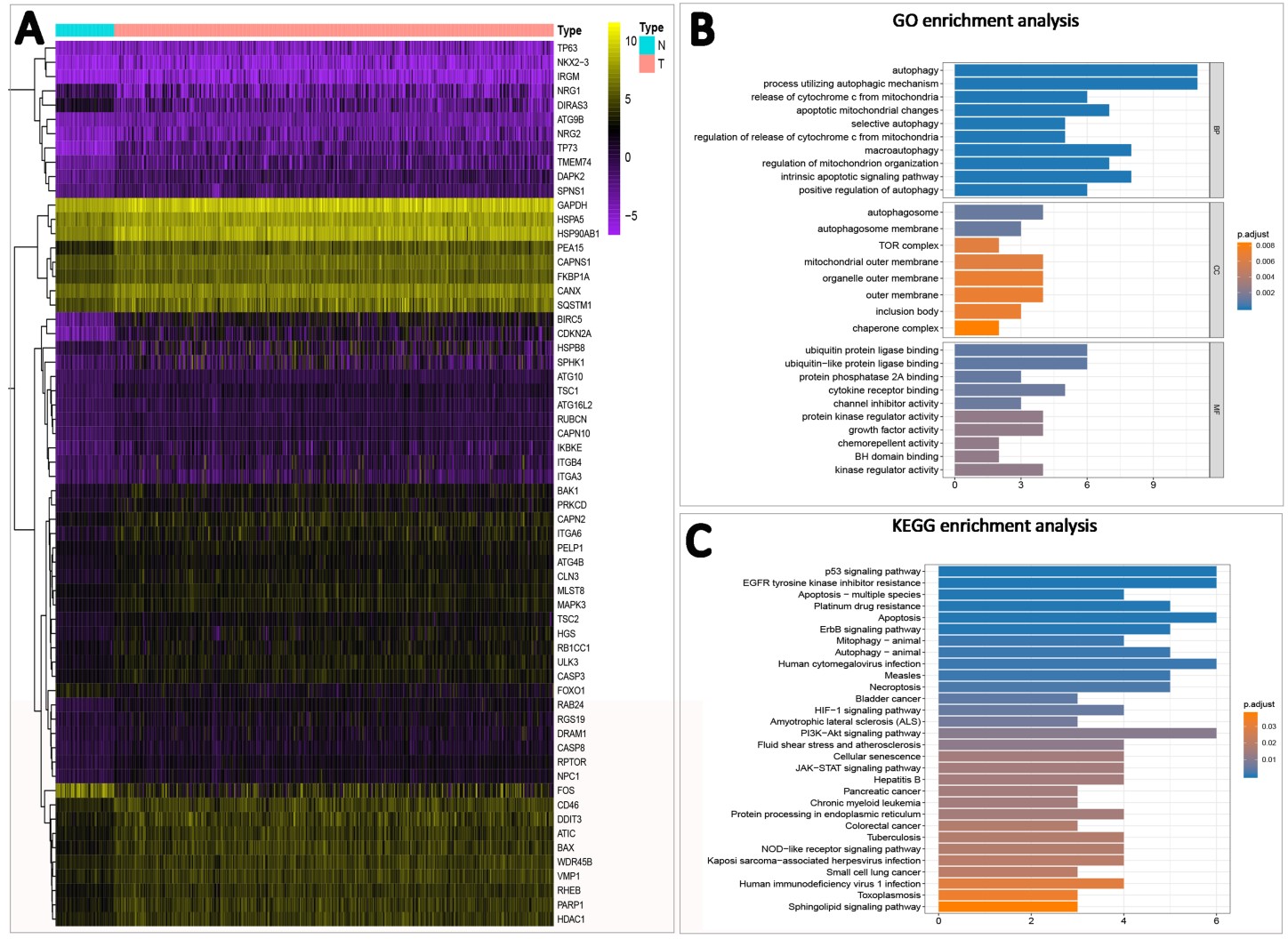

**Figure 1 Autophagy-related gene screening and enrichment analysis.** (A) Sixty-three differentially expressed genes of HCC in the TCGA dataset. N represents normal tissue and T represents tumor tissue. (B and C) Differential gene GO and KEGG enrichment analysis, BP stands for biological process, CC stands for cellular component, MF stands for molecular function.

concentrated in the PI3K–AKT signaling pathway, Platinum drug resistance, Hepatitis B (Figs. 1B and 1C).

## Prognostic value of ATGs and a risk signature built using seven selected ATGs

To investigate the prognostic role of ATGs in HCC. Univariate Cox regression analysis was performed to explore the expression levels in the TCGA dataset. The results showed that 21 genes are significantly correlated with OS ($P < 0.001$). All 21 genes are risky genes with Hazard ratio >1 (Fig. 2A). To better predict the clinical pathological features and prognosis of HCC with ATGs, the least absolute shrinkage and selection operator (LASSO) Cox regression algorithm was applied to the 21 prognosis-associated genes in the TCGA dataset, which was used as a training set. Seven ATGs were selected to build the risk

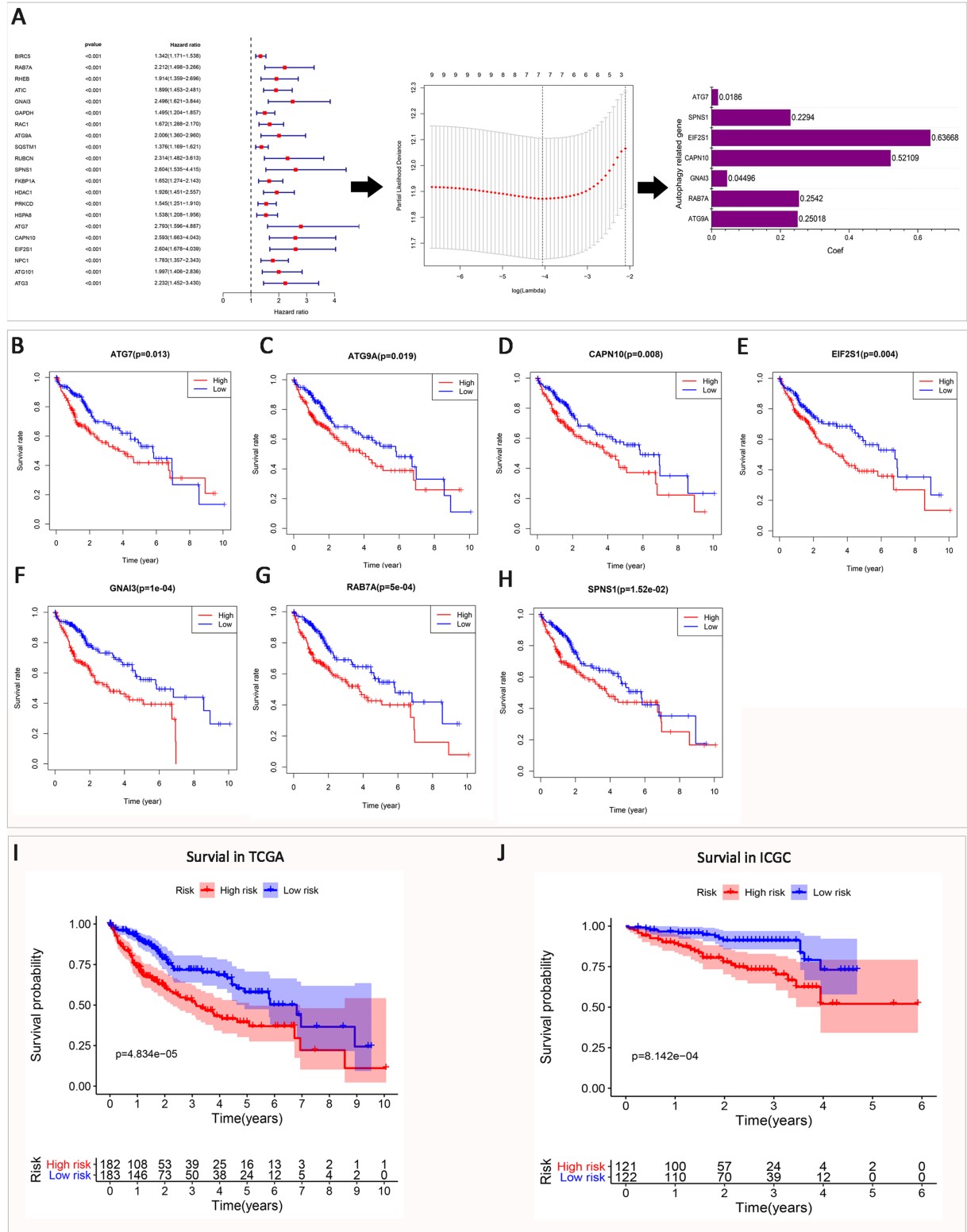

**Figure 2 Establishment of risk signature with seven ATGs.** (A) The process of building the risk signature containing seven ATGs. (B–H) Kaplan–Meier OS curve in the TCGA dataset. The seven ATGs which is used to build the risk signature affects the prognosis of HCC patients. (I and J) OS curves for patients between high risk and low risk in the TCGA and ICGC datasets.

signature based on the minimum partial Likelihood Deviance and the coefficients were used to calculate the risk score for both the TCGA and ICGC datasets. At the same time, we also analyzed the prognostic value and clinical pathology for seven ATGs which were used to build the risk signature in the TCGA dataset (Figs. 2B–2H; Figs. S1 and S2). These ATGs affect OS in HCC patients. Based on the median risk value, we divided the TCGA and ICGC dataset into a high-risk group and a low-risk group. The survival time of the high-risk group was significantly lower than that of the low-risk group ($P < 0.001$) (Figs. 2I and 2J). This indicates that the risk score has an important predictive value for the prognosis of HCC (Fig. S3). Seven ATGs were differentially expressed in the high and low risk group ($P < 0.0001$) (Fig. S4).

## Prognostic risk score showed a correlation with clinicopathological features of HCC

Univariate and multivariate Cox regression analyses were used to observe whether risk signature is an independent prognostic indicator for HCC. Univariate analysis showed that stage and risk score was associated with OS in patients with HCC ($P < 0.001$, Fig. 3A). We put the clinicopathological features and risk score into multivariate Cox regression. The results showed that the stage and risk score were still closely related to the OS of patients ($P < 0.001$, Fig. 3A). In the validation of the ICGC dataset, we get the same result. Stage and risk scores are closely related to overall patient survival in univariate and multivariate Cox regression analyses. Besides, we also found that gender is associated with OS in the ICGC dataset ($P < 0.05$, Fig. 3B). In summary, through the above analysis, we fully determined that the risk score from seven ATGs can independently predict the prognosis of HCC.

The ROC curve showed that the risk score (AUC = 0.755) was a perfect predictor of the 5 year survival rate of HCC patients in TCGA dataset (Fig. 3C), and the predicted effect was higher than grade (AUC = 0.505), stage (AUC = 0.663), age (AUC = 0.535), gender (AUC = 0.513). In the validation of the ICGC dataset, we found that the risk score can also predict the patient's 5 year survival rate (AUC = 0.698) (Figs. 3C and 3D).

In the front, univariate and multivariate Cox regression analyses showed that stage and risk scores were independent prognostic factors affecting the OS of patients with HCC. Further analysis showed that the risk score was closely related to clinicopathological features of HCC patients. As the stage state of HCC patients increases, the risk score increases accordingly in TCGA and ICGC datasets (Figs. 3E and 3F; Fig. S5). The above results fully prove that the prognostic risk signature is closely related to the malignant degree of tumor. The higher the degree of malignancy, the stronger the autophagy. Finally, we performed a gene set enrichment analysis (GSEA) on the high-risk group to show that immune-related hallmarks were enriched (Figs. 3G–3J). Including PI3K–AKT–mTOR signaling, TGF-BETA signaling, IL6-JAK-STAT3 signaling, IL2-STAT5 signaling. The results of the above enrichment analysis suggest that the prognosis signature of ATGs may be closely related to the immune response.

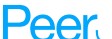

**Figure 3 Correlation between prognostic signature and clinicopathological features of HCC.** (A and B) Clinicopathological features and OS of Univariate and multivariate Cox regression analyses in the TCGA and ICGC datasets. (C and D) The ROC curve shows that the risk score predicts the 5 year survival rate of patients with HCC. (E and F) The higher the stage of HCC patients, the higher the risk score. (G–J) GSEA showed that genes with higher expression in high-risk group were enriched for Hallmarks of the immune response.

## Prognostic signatures of ATGs are closely related to PI3K–AKT–mTOR signaling

In the previous, we found through GSEA that high-expression genes in the high-risk group were mainly enriched in PI3K–AKT–mTOR signaling (NES = 2.102, Nom $P < 0.001$, FDR $q = 0.004$). Therefore, here we analyzed the expression of PI3K–AKT–mTOR signaling-related proteins in different risk groups through the TCPA and TCGA databases, and the results showed that AKT and mTOR protein (Figs. 4A–4F) and mRNA (Figs. 4G–4I) expression were up-regulated, but phosphorylated AKT (AKT–pS473, $P = 0.003$) (Fig. 4B) and mTOR (mTOR–pS2448, $P = 0.0116$) (Fig. 4D) expression down-regulated, indicating inhibition of AKT and mTOR protein phosphorylation in the high-risk group, resulting in inhibition of the PI3K–AKT–mTOR signaling pathway. Meanwhile, we found that mTOR target molecule P70S6K1 protein and mRNA expression were up-regulated, while P70S6K–pT389 ($P < 0.0001$) expression was down-regulated. It is well demonstrated that ATGs promote HCC progression by inhibiting PI3K–AKT–mTOR signaling. Finally, we analyzed the OS of mTOR–pS2448 in different risk groups, and the survival time of high expression of mTOR–pS2448 in the high-risk group was significantly higher than that in the low-expression group ($P = 0.045$). However, in the low-risk group, the mTOR–pS2448 expression had no significant effect on the prognosis of patients (Figs. 4J and 4K). Also, mTOR mRNA shows the same phenomenon (Fig. S6). The phosphorylation of mTOR is inhibited during high-intensity autophagy, so the survival time of mTOR–pS2448 with low expression is shortened. Combined with the effect of mTOR–pS2448 on the OS of HCC patients, we found that autophagy affects malignant tumor progression and is closely related to PI3K–AKT–mTOR signaling.

## Differences of biomarkers predicted by immune cell infiltration and immunotherapy among different risk groups

To explore the differences in immune cell infiltration between different risk groups, we used the CIBERSORT and TIMER methods to assess the abundance of each sample based on the TCGA dataset. The results showed that the infiltration level of macrophages in the high-risk group was higher than that in the low-risk group (Figs. 5A–5D). Using CIBERSORT for the ICGC dataset we come to the same conclusion (Fig. S7). At the same time, CD11B mRNA and F4/80 mRNA expression were found to be significantly different in different risk groups based on the TCGA dataset. CD11B and F4/80 mRNA are characteristic molecular markers on the surface of macrophages, which can indirectly predict macrophage abundance (*Maruyama et al., 2005*; *Schledzewski et al., 2006*).

PD-L1 and CTLA-4 have been shown to be targeted for a variety of tumor immunotherapy (*Chae et al., 2018*; *Ott, Hodi & Robert, 2013*), In HCC, Phase I clinical trials have shown that PD-L1 and CTLA-4 blockers have achieved good clinical results (*Kudo, 2017*). In this study, we observed significant differences in PD-L1 and CTLA-4 mRNA expression between different risk groups (Figs. 5E, 5F, 5H and 5I). In the high-risk group, PD-L1 and CTLA-4 mRNA expression were up-regulated in the TCGA and ICGC datasets. At the same time, we also noticed that the PD-L1 protein was also significantly elevated in the high-risk group based on the TCPA database (Fig. 5G). In addition, PD-L1

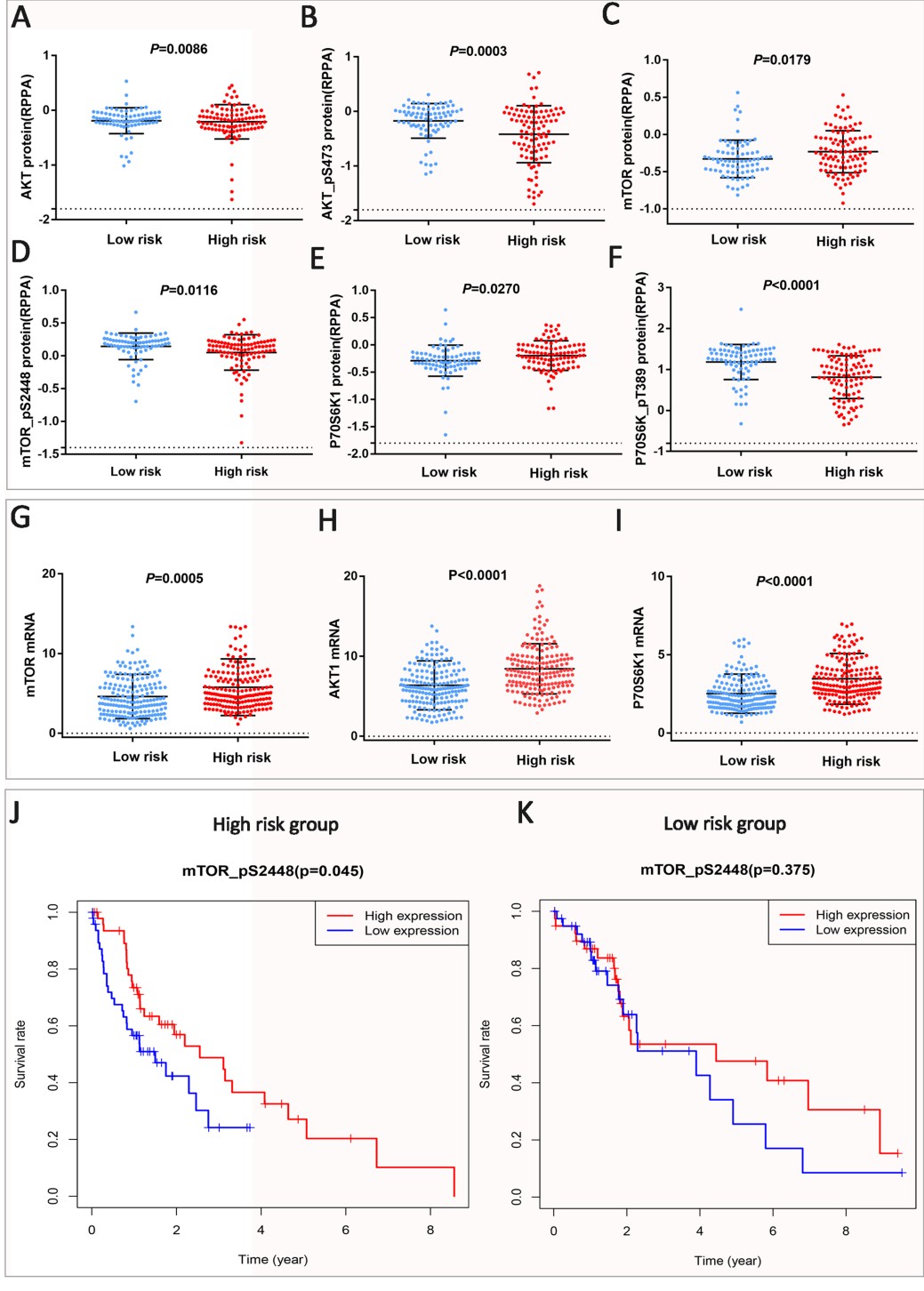

**Figure 4 Autophagy promotes HCC progression by inhibiting PI3K–AKT–mTOR signaling.** (A–F) The expression of PI3K–AKT–mTOR signaling related protein is based on TCPA database. (G–I) PI3K–AKT–mTOR signaling-related mRNA expression in the TCGA datasets. (J and K) Kaplan–Meier OS curve of HCC patients is shown. mTOR–pS2448 protein high expression patients survived shorter than low expression patients in the high-risk group; in the low-risk group, mTOR–pS2448 protein expression was not significantly different from patient prognosis.

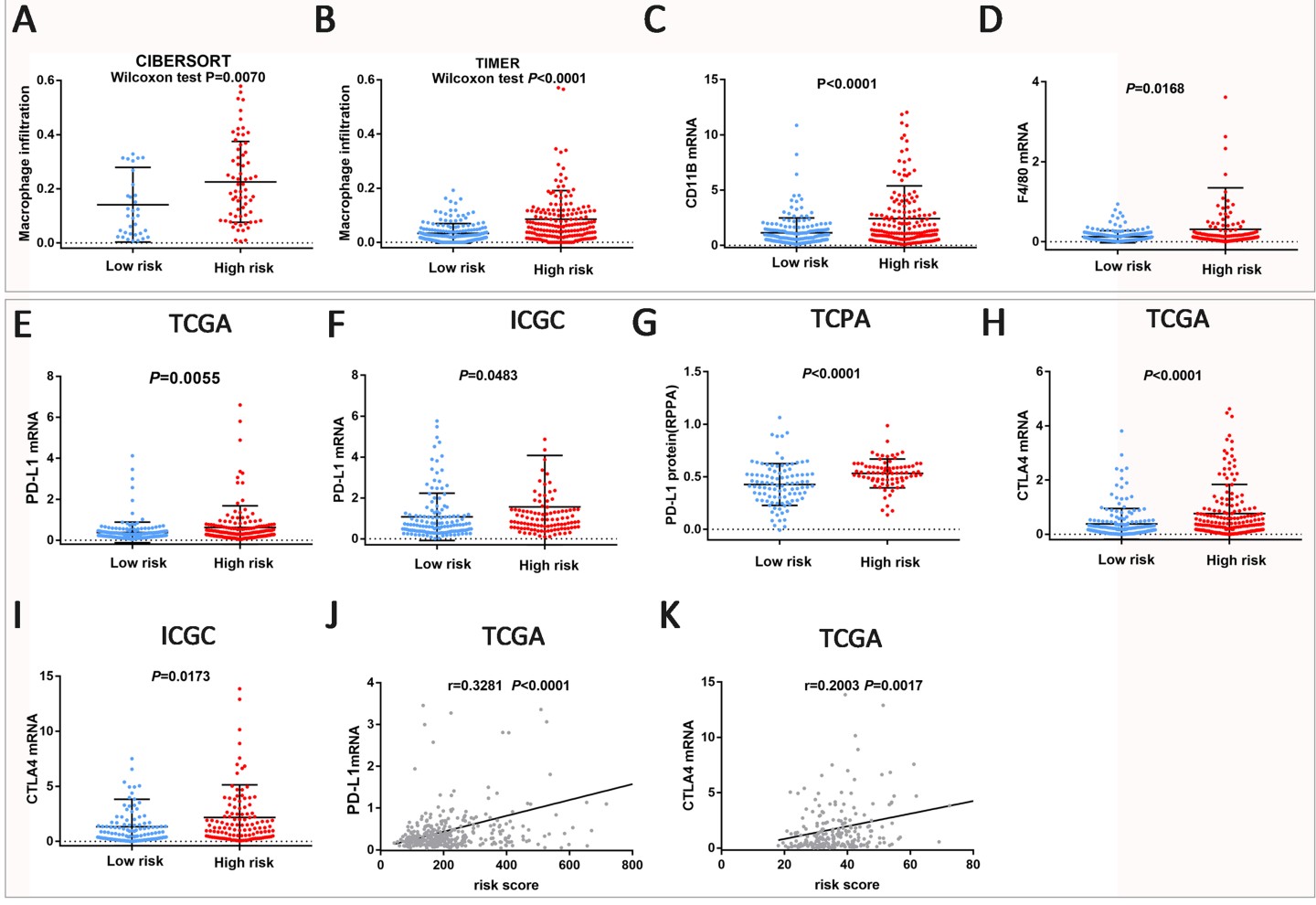

**Figure 5 Biomarkers predicted by immune cell infiltration and immunotherapy among different risk groups.** (A–D) Abundance of Macrophage infiltration was quantified with TIMER or CIBERSORT and macrophage marker expression in the TCGA dataset. (E–K) Difference and correlation of PD-L1 and CTLA-4 expression in different risk groups in the TCGA and ICGC datasets.

and CTLA-4 mRNA expression levels showed a significant positive correlation with the risk score in the TCGA dataset (Figs. 5J and 5K). The above results indicate that the risk signature can be used as a predictive marker for immunotherapy response.

## DISCUSSION

Recurrence is easy after the operation, and chemotherapy resistance has become the main factor of the higher death rate of HCC. ATGs have been confirmed to affect tumor progression and prognosis in breast cancer (*Yao et al., 2011*), lung cancer (*Xu et al., 2011*), colorectal cancer (*Bednarczyk et al., 2017*), bladder cancer (*Eissa et al., 2017*) and other tumors. In this study, we confirmed that ATGs are closely related to the clinicopathological features and prognosis of HCC. By analyzing the expression of tumor tissue and normal tissues, we identified 63 differentially expressed ATGs. GO and KEGG enrichment analysis was mainly enriched in autophagy, PI3K–AKT–mTOR signaling pathway, Platinum drug resistance. Based on the LASSO Cox regression algorithm, seven ATGs

(ATG9A, RAB7A, GNAI3, CAPN10, EIF2S1, SPNS1 and ATG7) were selected to establish prognostic risk signatures, and HCC patients were divided into low-risk and high-risk groups. Kaplan–Meier method analysis found that the survival time of the high-risk group was significantly shorter than that of the low-risk group. In addition, we also demonstrated that seven ATGs are closely related to the OS of HCC. It has been previously reported that ATG7 and ATG9 key regulatory factors affect the prognosis and progression of gastric cancer (*Cao et al., 2016*). Cell migration is dependent on Rab7a expression, and Rab7a provides a directional effect on vimentin (*Margiotta et al., 2017*). The high expression of CAPN10 suggests a good prognosis for esophageal squamous cell carcinoma (*Chan et al., 2013*), however, the high expression of CAPN10 in this study suggests a poor prognosis for HCC. These findings indicate that up-regulated ATGs affect multi-system tumor progression and prognosis, and have different mechanisms of action in different tumors.

In this section, we analyzed the risk score and stage as an independent prognostic indicator for HCC and confirmed it in the validation dataset ICGC. Further analysis revealed that the risk score was also increased in the high stage state. It indicates that the prognosis risk signature is an indicator of poor prognosis of HCC, suggesting that ATGs promote HCC to malignant transformation. Next, GSEA for the high-risk group showed that these highly expressed genes were associated with immune responses and PI3K–AKT–mTOR signaling. Previous studies have shown that autophagy promotes tumor progression by inhibiting mTOR signaling (*Dunlop & Tee, 2014*; *Jung et al., 2010*). The results of this study are consistent with those of others. By comparing the expression of PI3K–AKT–mTOR signaling pathway-related proteins in different risk groups, the expression of AKT and mTOR proteins is up-regulated but the expression of phosphorylated AKT (AKT–pS473) and mTOR (mTOR–pS2448) is down-regulated, which indicates that PI3K–AKT–mTOR signaling is inhibited in high-risk groups and proves that mTOR–pS2448 affects the prognosis of HCC patients in the high-risk group.

*Baghdadi et al. (2013)* reported that activation of phagocyte APKα1 signal by TIM-4 enhanced the autophagy process of cells, leading to degradation of tumor antigen and reduction of cytotoxic T lymphocyte reaction, thus mediating chemotherapy resistance. Hence, we highly suspect that HCC ATGs are closely related to tumor immune cells and immune genes. M2 macrophages play a major role in tumor and produce epidermal growth factor, vascular endothelial growth factor and ornithine to promote tumor cell growth, while M1 macrophages depress tumor growth by secreting IL-12 and IFN-γ. Increasing the proportion of M1 macrophages through macrophage transformation has an anti-tumor effect (*Mills, Lenz & Harris, 2016*). The benefits of PD-L1 inhibitors to patients are generally recognized. PD-L1 expression is the main predictor of the PD-L1 therapeutic effect (*Shukuya & Carbone, 2016*). *Chen et al. (2012)* found that the expression of PD-L1 in HCC is related to the inflammatory microenvironment in which macrophages participate. We found that the prognosis signature of ATGs is positively correlated with macrophage infiltration and PD-L1 expression. Thus, the autophagy process affects HCC macrophage infiltration, and the prognosis signature of ATGs can predict the PD-L1 therapeutic effect. And studies show that the combination of autophagy blocker and

PD-L1 inhibitor has a significant anti-tumor effect compared with PD-L1 inhibitor alone (*Robainas et al., 2017*).

## CONCLUSION

We found that ATGs are differentially expressed in HCC, and proved the value of ATG prognostic signature in predicting the OS of HCC. Further analysis shows that autophagy promotes the progression of HCC by inhibiting mTOR signaling. Finally, the prognosis risk signature of ATGs is positively correlated with immune cell infiltration and the known immunotherapeutic marker PD-L1. The prognosis risk signature of ATGs can be as potential predictive markers for immunity in HCC.

## ABBREVIATIONS

| | |
|---|---|
| **ATGs** | Autophagy-related genes |
| **HCC** | Hepatocellular carcinoma |
| **TCPA** | The Cancer Proteome Atlas |
| **TCGA** | The Cancer Genome Atlas |
| **ICGC** | International Cancer Genome Consortium |
| **GO** | Gene Ontology |
| **KEGG** | Kyoto Encyclopedia of Genes and Genomes |
| **GSEA** | Gene set enrichment analysis |
| **TIMER** | Tumor IMmune Estimation Resource |
| **David** | The Database for Annotation, Visualization and Integrated Discovery |
| **ROC** | Receiver operating characteristics |
| **OS** | overall survival |
| **LASSO** | The least absolute shrinkage and selection operator. |

### Funding

This study was supported by grants from the National Science Foundation of China (NSFC, No. 31770985), the Social Development Project of Jiangsu Province (No. BE2019665), the Postdoctoral Science Foundation Grant of China (No. 2016M591913), the Jiangsu Provincial Medical Youth Talent (No. QNRC2016732), the Jiangsu Provincial "Six Peaks Talent" Program (No. 2016-WSW-043), the Suzhou Municipal Project of Gusu Health Talent, Young Top Talent (No. 2018-057), the Gusu Health Talents Cultivation Program (No. GSWS2019028), the Scientific Research Program of Jiangsu Provincial "333 Projects" (No. BRA2019327), the Science and Technology Program of Suzhou City, China (Nos. SYS2019053 and SLC201906), the Provincial Natural Science Foundation of Jiangsu Province (No. BK20161225), and the Scientific Research Program of Jiangsu Provincial Commission of Health and Family Planning (No. H201620). The funders had no role in study design, data collection and analysis, decision to publish, or preparation of the manuscript.

## Grant Disclosures

The following grant information was disclosed by the authors:
National Science Foundation of China (NSFC): 31770985.
Social Development Project of Jiangsu Province: BE2019665.
Postdoctoral Science Foundation Grant of China: 2016M591913.
Jiangsu Provincial Medical Youth Talent: QNRC2016732.
Jiangsu Provincial "Six Peaks Talent" Program: 2016-WSW-043.
Suzhou Municipal Project of Gusu Health Talent, Young Top Talent: 2018-057.
Gusu Health Talents Cultivation Program: GSWS2019028.
Scientific Research Program of Jiangsu Provincial "333 Projects": BRA2019327.
Science and Technology Program of Suzhou City, China: SYS2019053 and SLC201906.
Provincial Natural Science Foundation of Jiangsu Province: BK20161225.
Scientific Research Program of Jiangsu Provincial Commission of Health and Family Planning: H201620.

## Competing Interests

The authors declare that they have no competing interests.

## Author Contributions

- Deli Mao conceived and designed the experiments, performed the experiments, analyzed the data, prepared figures and/or tables, authored or reviewed drafts of the paper, and approved the final draft.
- Zhe Zhang performed the experiments, prepared figures and/or tables, and approved the final draft.
- Xin Zhao conceived and designed the experiments, prepared figures and/or tables, and approved the final draft.
- Xiaoqiang Dong conceived and designed the experiments, authored or reviewed drafts of the paper, and approved the final draft.

## Data Availability

The gene expression profile and the clinical and pathological information that support the findings of this study are available in The Cancer Genome Atlas (https://portal.gdc.cancer.gov/).

We selected TCGA-LIHC specimens from patients with liver cancer, and Data Category selected transcriptome profiling; Workflow Type selects HTSeq-FPKM.

The information of liver cancer patients are available in the International Cancer Genome Consortium (ICGC: https://dcc.icgc.org/releases/current/Projects/LIRI-JP) and The Cancer Proteome Atlas database (TCPA: https://tcpaportal.org/tcpa/download.html).

## Supplemental Information

Supplemental information for this article can be found online at http://dx.doi.org/10.7717/peerj.8383#supplemental-information.

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
