# Peer review of "Autophagy-related genes prognosis signature as potential predictive markers for immunotherapy in hepatocellular carcinoma"

_PeerJ, doi:10.7717/peerj.8383_

## Round 0.1 · original submission · Major Revisions

Please address all the critiques of the reviewers and amend your manuscript accordingly

Reviewer 1 ·

Basic reporting

In this manuscript, the authors demonstrated that many Autophagy-related genes are differentially expressed in hepatocellular carcinoma (HCC). The authors further showed the potential involvement of PI3K-AKT-mTOR signaling pathway candidates in HCC and concluded that the prognostic effects of ATGs could be useful.

Experimental design

All experiments were designed clearly.

Validity of the findings

The results of the bioinformatic studies were well stated. However, most of the findings were either well-expected or well known in the field. May be, supplementation of these findings with in vitro/in vivo studies could be interesting and useful for the manuscript.

Additional comments

This manuscript looked at the potential use of ATGs as a marker for Hepatocellular carcinoma. The role of autophagy in HCC is already reported and well documented. Previous studies clearly established the inhibitory effect of ATGs on PI3K-AKT-mTOR signaling, which in turn may promote tumor progression. In this study, analyzing RNA-seq transcriptome profiles of HCC patients unsurprisingly enriched multiple differentially expressed genes in the ATG family. This enrichment is well expected from the already published results and unfortunately lacks novelty. However, the differential investigation of biomarkers using immune cell infiltration is relatively new and required to be investigated more, especially in today's world. My suggestion to the authors is to either concentrate on the novel part of the manuscript and try to build a story on it or to follow up on the claims within vitro/in vivo results.

Reviewer 2 ·

Basic reporting

Mao et al. characterized differentially expression of multiple autophagy related genes in tumor and normal tissue, and proposed the expression can be used as prognostic marker for certain liver cancer. The authors further showed that these upregulation of ATG is related to immune cell infiltration, which provide a mechanism basis for cancer immunotherapy strategies. In principle, this is an interesting study.

1, Fig.1A, In x-axis, what are the color codes cyan and red stand for? Is data regarding gene expression of normal tissue also included?
2, Fig. 1B, the GO and KEGG data is not showing in a standard fashion. Please modify it to show a bar plot with p-value.
3, Page 12, lane 124, the figure legend should not be inserted into the main text.

Experimental design

4, The work is largely based on the bioinformatic analysis. To ensure the reproducibility, the authors should include the original code of all the bioinformatic analysis in the supplemental data.

Validity of the findings

5, Page 15, lane 165. The authors wrote that “the above results fully demonstrate that the prognosis risk signature is the driving factor for the malignant progression of the tumor”, which is overstatement of the data . Actually, all analysis showing here can only provide a correlation between tumor malignancy and these markers.
6, Page 15, lane 173-174. I disagree the title used here “Autophagy-related genes promote HCC progression by inhibiting PI3K-AKT-mTOR signaling. As discussed above, the analysis itself doesn’t support this claim. Please modify the title as well as the conclusion in lane 192-193, “we further confirmed that autophagy promotes the malignant process of HCC by inhibiting PI3K-AKT-mTOR signaling.”

---

## Round 0.2 · accepted · Accept

Since all the critiques were adequately addressed and the manuscript was amended accordingly, the revised version is acceptable now.

Reviewer 2 ·

Basic reporting

The authors clearly reported the study.

Experimental design

The experiment is well designed.

Validity of the findings

All the original codes is now provided.

Additional comments

The authors have well addressed my concerns.